# Scalable Distributional Robustness in a Class of Non Convex Optimization with Guarantees

**Avinandan Bose**
University of Washington
avibose@cs.washington.edu

**Arunesh Sinha**
Rutgers University
arunesh.sinha@rutgers.edu

**Tien Mai**
Singapore Management University
atmai@smu.edu.sg

## Abstract

Distributionally robust optimization (DRO) has shown lot of promise in providing robustness in learning as well as sample based optimization problems. We endeavor to provide DRO solutions for a class of sum of fractionals, non-convex optimization which is used for decision making in prominent areas such as facility location and security games. In contrast to previous work, we find it more tractable to optimize the equivalent variance regularized form of DRO rather than the minimax form. We transform the variance regularized form to a mixed-integer second order cone program (MISOCP), which, while *guaranteeing near global optimality*, does not scale enough to solve problems with real world data-sets. We further propose two abstraction approaches based on clustering and stratified sampling to increase scalability, which we then use for real world data-sets. Importantly, we provide near global optimality guarantees for our approach and show experimentally that our solution quality is better than the locally optimal ones achieved by state-of-the-art gradient-based methods. We experimentally compare our different approaches and baselines, and reveal nuanced properties of a DRO solution.

## 1 Introduction

Distributionally robust optimization (DRO) is a popular approach employed in robust machine learning. Mostly, if not always, these works have focussed on the task of classification or regression. However, often in practical applications the end goal of learning is a decision output $\mathbf{z}$, which requires yet another complex optimization that uses the output $\widehat{\mathbf{x}}_1, \ldots, \widehat{\mathbf{x}}_N$ of a regressor $f(\cdot)$. For example, in facility location problem the learning of facility values is followed by an optimization using the values predicted to decide where to locate facilities and in security games adversary behavior model is learned and then an optimal defense allocation computed based on the learned model. Often the learning output is provided as public datasets with no access to the underlying private dataset used for such learning. In this set-up, we aim to provide robustness at the decision making level with access to only the non-robust learning output $\widehat{\mathbf{x}}_1, \ldots, \widehat{\mathbf{x}}_N$.

However, often the objective $F(\mathbf{z}, \mathbf{x})$ of decision optimization is *non-convex* in the learning system output $\mathbf{x}$ unlike the convex objective of classification or regression, presenting significant scalability challenges. In general, for decision making and specifically for the problem domains we consider, *global optimality is important* as sub-optimal decisions can lead to large revenue loss or loss of life; thus, the local optimality provided by gradient based methods is not sufficient. As a consequence, in this paper, we study the scenario of calculating DRO decisions using the given multi-dimensional real valued outputs $\widehat{\mathbf{x}}_1, \ldots, \widehat{\mathbf{x}}_N$ of a non-robust learned $f$. A first result (Theorem 1) characterizes the

36th Conference on Neural Information Processing Systems (NeurIPS 2022).

quality of DRO decision output compared to the scenario where we know the true $f^*$. Our main focus is on addressing the scalability issue for the DRO decision making problem for a particular, but widely used, class of *sum of fractionals non-convex objective*. This objective arises from the well-known *discrete choice models* [Train, 2003] of human behavior, which is known to *not* have scalable globally optimal solutions [Schaible, 1995, Li et al., 2019]; we use this for tackling two different decision optimization problem in facility location and a robust version of Bayesian Stackelberg security games problem with quantal response. As far as we know, this is a first attempt to solve the aforementioned non-convex problems in a DRO setting to near global optimality.

Our *first contribution* is a modelling construct, where we reformualate the variance regularized form [Duchi and Namkoong, 2019] of our non-convex sum of fractionals objective as a mixed integer second order cone program (MISOCP). While the MISOCP form provides more scalability than the original formulation and guaranteed solution quality (Theorem 2), it still does not scale to real world sized datasets. Our *second contribution* is a pair of approaches that achieves further scalability by splitting the problem space into sub-regions and solving a smaller MISOCP over representative samples from the sub-regions. Under mild conditions, both approaches provide *global optimality guarantees* (Theorem 3, 4).

Our *final contribution* is detailed experiments validating the scalability of our approaches on a simulated security game problem as well as two variants of facility location using park and ride data-sets from New York [Holguin-Veras et al., 2012]. We compare with two gradient-based approaches [Lin et al., 2020] and show the superior solution quality achieved by our approach, which also reveals the need for global optimality. We further show a nuanced property of the DRO solution in providing better decisions for low probability scenarios over non-robust versions. Overall, our work provides desired *robustness with globally optimal solution guarantees*.

**Related work:** Our work is built on a recent line of research that connects the concepts of DRO and variance regularization [Duchi and Namkoong, 2019, Duchi et al., 2021, Lam, 2016, Maurer and Pontil, 2009, Staib et al., 2019]. While most the previous studies along this research line focus on convex and continuous problems or problems with submodular objectives, our work concerns a class of DRO problems with fractional structures, which are highly non-convex and requires new technical developments for globally optimal solution. Recent work Yan et al. [2020], Qi et al. [2021] has addressed non-convex objectives in DRO using gradient based methods that converge to stationary points, which is insufficient for decision making as we experimentally show that stationary points and globally optimal points can yield very different decision utilities.

The literature on DRO is vast and we refer the reader to Rahimian and Mehrotra [2019] for a review. DRO methods can be classified by different ways to define ambiguity sets of distributions, for instance, ambiguity sets based on $\phi$-divergences [Ben-Tal et al., 2013, Duchi and Namkoong, 2019, Staib et al., 2019] or Wasserstein distances [Pflug and Wozabal, 2007, Esfahani and Kuhn, 2018, Shafieezadeh-Abadeh et al., 2015, Blanchet and Murthy, 2019]. In this work, we focus on $\phi$-divergence based models, motivated by their interesting connections with variance regularization and the tractability of the resulting non-convex DRO models.

We show that our DRO methods can be used in some popular decision-making problems such as Stackelberg security game (SSG) with Quantal Response [Tambe, 2011, Xu, 2016, Fang et al., 2017, Sinha et al., 2018, Yang et al., 2012, Haghtalab et al., 2016] or competitive facility location under random utilities [Benati and Hansen, 2002, Freire et al., 2016, Mai and Lodi, 2020, Dam et al., 2021]. To the best of our knowledge, a DRO Bayesian model has not been studied in existing SSG works. In the context of competitive facility location under random utilities, we seem to be the first to bring DRO as a consideration. We handle a DRO version of a facility cost optimization problem, which has also never been studied in prior work.

## 2   Background, Preliminary Notation and Result

We use bold fonts for vectors and non-bold font for vector components and scalars, e.g., $x_j$ is a component of $\mathbf{x}$. $[N]$ denotes $\{1, \ldots, N\}$. A $d$-dimensional vector is written as $\mathbf{x} = (x_j)_{j \in [d]}$ or as $(x_1, \ldots, x_d)$. The positive part of a vector $\mathbf{x}$ is $\mathbf{x}^+ = (\max(0, x_j))_{j \in [d]}$, and the negative part is $\mathbf{x}^- = (\min(0, x_j))_{j \in [d]}$. $\mathbf{0}, \mathbf{1}$ represent the all zero and all one vector.

**Distributionally Robust Optimization**: Consider a function $F$ with inputs being a decision variable $\mathbf{z}$ and parameter $\mathbf{x} \in X$. Both $\mathbf{z}$ and $\mathbf{x}$ lie in an Euclidean space and both *are constrained by linear constraints*; for notational ease we skip writing the constraints in the general formulation. We seek to maximize the following objective function $\max_{\mathbf{z}} \mathbb{E}_P[F(\mathbf{z}, \mathbf{x})]$, where $\mathbf{x}$ is distributed according to $P$. The details of how $P$ arises from an underlying regression problem is stated later in the text just before Theorem 1. For many classes of distributions the above is generally not tractable and one needs to sample $\mathbf{x}$ from $P$. Let $\widehat{\mathbf{x}}_1, \ldots, \widehat{\mathbf{x}}_N$ be $N$ samples, we can solve the sample average approximation (SAA) problem instead $\max_{\mathbf{z}} \frac{1}{N} \sum_{n \in [N]} F(\mathbf{z}, \widehat{\mathbf{x}}_n)$. Let $\widehat{P}_N$ be the empirical distribution induced by the samples. The SAA above is same as $\max_{\mathbf{z}} \mathbb{E}_{\widehat{P}_N}[F(\mathbf{z}, \mathbf{x})]$. A distributionally robust version of the SAA problem is $\max_{\mathbf{z}} \min_{\widetilde{P} \in \mathcal{P}_{\xi,N}} \left\{ \mathbb{E}_{\widetilde{P}}[F(\mathbf{z}, \mathbf{x})] \right\}$, where the ambiguity set $\mathcal{P}_{\xi,N} = \left\{ \widetilde{P} \big| \mathcal{D}_\phi(\widetilde{P} || \widehat{P}_N) \leq \xi/N \right\}$, and $\mathcal{D}_\phi(P||Q)$ is the $\chi^2$ divergence: $\mathcal{D}_\phi(P||Q) = \frac{1}{2} \int (dP/dQ - 1)^2 dQ$. The above optimization can be written equivalently as ($\Delta_{\xi,N}$ defined below)

$$\max_{\mathbf{z}} \min_{\mathbf{p} \in \Delta_{\xi,n}} \left\{ \sum_{i \in [N]} p_i F(\mathbf{z}, \widehat{\mathbf{x}}_i) \right\} \tag{DRO}$$

where $\Delta_{\xi,N} = \left\{ \mathbf{p} \in \mathbb{R}_+^N \big| \sum_i p_i = 1; \ ||\mathbf{p} - \mathbf{1}/N||_2^2 \leq 2 \frac{\xi}{N^2} \right\}$. We have earlier stated that $\widehat{\mathbf{x}}_i$ is output by a regressor, say $f \in \mathcal{F}$ for some function class $\mathcal{F}$ trained using loss $\mathcal{L}$ with $N_T$ datapoints, but this implies that $\widehat{\mathbf{x}}_i = f(b_i)$ might not exactly same as $\mathbf{x}_i^* = f^*(b_i)$ for some underlying feature values $b_i$ and best function $f^* \in \mathcal{F}$. We assume $f^*$ is deterministic and has zero Bayes risk.

Let $D$ be the probability distribution from which the feature values $b_i$ are sampled. Then, let $P^*$ be the true distribution on $X$ induced by $f^*$ acting on the feature values that are distributed according to $D$ (i.e., the pushforward measure). Similarly, $P$ is the distribution on $X$ induced by $f$ acting on the feature values that are distributed according to $D$. Thus, the (unknown) samples $\mathbf{x}^*$'s are obtained from $P^*$; hence, the true utility of any decision $\mathbf{z}$ is $\mathbb{E}_{P^*}[F(\mathbf{z}, \mathbf{x})]$. We prove an end to end guarantee about the output decision $\widehat{\mathbf{z}}^{**}$ using $\widehat{\mathbf{x}}_i$'s, which reveals that $\widehat{\mathbf{z}}^{**}$ is not much worse than the decision $\mathbf{z}^{**}$ that would be learned if $\mathbf{x}_i^*$'s would be available and used. The result shows that larger training data $N_T$ helps.

**Theorem 1.** *Let $\mathbf{x}_i^* = f^*(b_i)$ for true function $f^*$ and let $\widehat{\mathbf{x}}_i = f(b_i)$ for the learned empirical risk minimizer $f$. Suppose the optimal decision when solving* **DRO** *is $\mathbf{z}^{**}$ using $\mathbf{x}_i^*$'s and $\widehat{\mathbf{z}}^{**}$ using $\widehat{\mathbf{x}}_i$'s. Also, let $F$ be $\tau$-Lipschitz in $\mathbf{x}$, $X$ be bounded, and a scaled $\mathcal{L}$ upper bound $|| \cdot ||_2$ (i.e., $||\mathbf{x} - \mathbf{x}'||_2 \leq \max(k\mathcal{L}(\mathbf{x}, \mathbf{x}'), \epsilon)$ for constants $k, \epsilon$) then, the following holds with probability $1 - 2\delta - 2\delta_1$: $\mathbb{E}_{P^*}[F(\widehat{\mathbf{z}}^{**}, \mathbf{x})] \geq \mathbb{E}_{P^*}[F(\mathbf{z}^{**}, \mathbf{x})] - C/\sqrt{N} - (1 + 2\sqrt{\xi})\tau\epsilon - \epsilon_N - \epsilon_{N_T}$, where $\epsilon_K = C_1 \mathcal{R}_K(\mathcal{L} \circ \mathcal{F}) + C_2/\sqrt{K}$ and $\mathcal{R}_K$ is the Rademacher complexity with $K$ samples and $C, C_1, C_2$ are constants dependent on $\delta, \delta_1, \xi, k, \tau$.*

**Variance Regularizer**: As a large number of samples are needed for a low variance approximation of the true distribution, another proposed robust version of the SAA [Maurer and Pontil, 2009, Duchi and Namkoong, 2019] is to optimize the following variance-regularized (VR) objective function

$$\max_{\mathbf{z}} \left\{ \mathbb{E}_{\widehat{P}_N}[F(\mathbf{z}, \mathbf{x})] - C \sqrt{\frac{\text{Var}_{\widehat{P}_N}(F(\mathbf{z}, \mathbf{x}))}{N}} \right\}. \tag{VR}$$

The above allows to directly optimize the trade-off between bias and variance. In a fundamental result, Duchi and Namkoong [2019, Theorem 1] show that, with high probability, problem (VR) is *equivalent* to the problem (DRO). Further, Duchi and Namkoong [2019] argue for solving the DRO version of the problem for concave $F$ (note we are solving a maximization SAA problem) since concave $F$ results in concavity of $\min_{\mathbf{p} \in \Delta_{\xi,n}} \sum_{i \in [N]} p_i F(\mathbf{z}, \widehat{\mathbf{x}}_i)$, thus, the overall DRO problem is a concave maximization problem. In contrast, the objective in (VR) is not concave.

In this paper, our focus is on $F$ that is *not concave*, thus, the choice of DRO or variance regularized form is not obvious. For the class of functions $F$ that we analyze, we argue the variance regularized version is more promising as far as scalability for global optimality is concerned. We work with the assumption that the variance regularized form is equivalent to DRO, which holds under the mild condition shown in Equation (9) in Duchi and Namkoong [2019].

# 3 Towards a Globally Optimal Solution

In this section, we present results for a general class of non-concave functions $F$ that has a fractional form with non-linear numerator and denominator and that can be approximated by a linear fractional form with binary variables. Then, we show three *prominent* applications of our approach.

**Notation**: For ease of notation, we use shorthand to denote $F(\mathbf{z}, \widehat{\mathbf{x}}_i)$ by $F_i$ and $2\frac{\xi}{N^2}$ by $\rho$.

## 3.1 General Recipe to Form a MISOCP

We perform a sequence of variable and constraint transformations of (VR), leading to a MISOCP.

**Mixed Integer Concave Program**: The variance regularized objective in shorthand notation is:

$$\mathcal{G}(\mathbf{z}) = \sum_i \frac{F_i}{N} - \sqrt{\rho \sum_i \Big(\frac{\sum_{i'} F_{i'}}{N} - F_i\Big)^2} \tag{1}$$

We substitute $l_i = \frac{\sum_{i'} F_{i'}}{N} - F_i$ and $q = \frac{\sum_{i'} F_{i'}}{N}$ for all $i \in [N]$, such that $\sum_i l_i = 0$ and $F_i = q - l_i$. The objective in Equation 1 thus becomes $q - \sqrt{\rho \sum_i l_i^2}$ which is concave in the variables $q$ and $\{l_i\}$. We add the new constraints $\sum_i l_i = 0$ and $F_i = q - l_i$ for all $i \in [N]$. Note that, while the objective is now concave with above changes, we have pushed the non-convexity into the constraints $F_i - q + l_i = 0$ for all $i \in [N]$ that are added to the optimization.

If $F_i$ can be written (or approximated) as a fraction with affine numerator and denominator, we can convert the constraint $F_i - q + l_i = 0$ into a convex constraint, giving us an overall concave program. The conversion is explained next. Suppose $F_i$ can be written (or approximated) as $\frac{\mathbf{a}_i^T \mathbf{v} + b_i}{\mathbf{a}_i'^T \mathbf{v} + b_i'}$ where $\mathbf{v}$ represents *binary* variables after conversion ($\mathbf{v}$ completely replaces $\mathbf{z}$ and $\mathbf{a}_i, \mathbf{a}_i', b_i, b_i'$ are dependent on $\widehat{\mathbf{x}}_i$'s). Typically, such a linear fractional form is constructed by discretizing the arguments of the original non-linear functions in the numerator and denominator of $F$. Assume $\mathbf{v}$ is of dimension $d$; typically $d$ will depend on the number of pieces. Define $\mathbf{y}_i = \mathbf{v} t_i$ where $t_i = \frac{1}{\mathbf{a}_i'^T \mathbf{v} + b_i'}$. Then, we can (re)write the fractional form for $F_i$ as $F_i = \mathbf{a}_i^T \mathbf{y}_i + b_i t_i$. This yields the linear constraints below with the non-linearity now restricted to $\mathbf{y}_i = \mathbf{v} t_i$.

$$\sum_{i=1}^{N} l_i = 0 \tag{2}$$

$$\mathbf{a}_i^T \mathbf{y}_i + b_i t_i = q - l_i \qquad \forall i \in [N] \tag{3}$$

$$\mathbf{a}_i'^T \mathbf{y}_i + b_i' t_i - 1 = 0 \qquad \forall i \in [N] \tag{4}$$

We handle $\mathbf{y}_i = \mathbf{v} t_i$ using McCormick relaxation technique [McCormick, 1976]. Typically, McCormick relaxation is applied for bilinear terms that are the product of two continuous variables, in which case, it is an approximation. However, in our case since $\mathbf{v}$ is a binary vector variable, the McCormick relaxation yields an exact reformulation of the bilinear term. For applying McCormick technique, we need an upper and lower bound of $\mathbf{v}$ and $t_i$. Since $\mathbf{v}$ a vector of binary variables, we have lower bound $\mathbf{v}^L = \mathbf{0}$ and upper bound $\mathbf{v}^U = \mathbf{1}$. Similarly, $t_i^L = \frac{1}{(\mathbf{a}_i'^+)^T \mathbf{1} + b_i'}$ and $t_i^U = \frac{1}{(\mathbf{a}_i'^-)^T \mathbf{1} + b_i'}$ (recall superscript $+$ and $-$ indicate positive and negative part of a vector respectively). Further, it is assumed $t_i^U$ and $t_i^L$ exist. Note that these bounds are not variables but fixed constants that depend on the fixed parameters $\mathbf{a}_i, \mathbf{a}_i', b_i, b_i'$, hence these need to be computed just once. Using the upper and lower bounds of $\mathbf{v}$ and $t_i$ in McCormick technique we get:

$$\mathbf{y}_i - \mathbf{v} t_i^U \leq 0; \qquad \forall i \in [N] \tag{5}$$

$$\mathbf{y}_i - (\mathbf{1} t_i + \mathbf{v} t_i^L - \mathbf{1} t_i^L) \leq 0; \qquad \forall i \in [N] \tag{6}$$

$$-\mathbf{y}_i + (\mathbf{1} t_i + \mathbf{v} t_i^U - \mathbf{1} t_i^U) \leq 0; \quad \forall i \in [N] \tag{7}$$

$$-\mathbf{y}_i + \mathbf{v} t_i^L \leq 0; \qquad \forall i \in [N] \tag{8}$$

$$\mathbf{v} \in \{0,1\}^d \tag{9}$$

$$t_i^U \leq t_i \leq t_i^L; \qquad \forall i \in [N] \tag{10}$$

It is straightforward to check the above set of equations is equivalent to $\mathbf{y}_i = \mathbf{v} t_i$. With the changes, we obtain a mixed integer concave program (with all constraints linear). Next, while the above can be solved using branch and bound with general purpose convex solvers for intermediate problem, we show that a further transformation to a MISOCP is possible. Specialized SOCP's solvers provide much more scalability than a general purpose convex solvers [Bonami and Tramontani, 2015] and hence partially address the scalability challenge.

**Mixed Integer SOCP**: We transform further by introducing another variable $s$ to stand for $\sqrt{\rho \sum_i l_i^2}$. We use $\mathbf{r} = (s, q, (l_i)_{i \in [N]}, \mathbf{v}, (t_i)_{i \in [N]}, \mathbf{y}_1, \ldots, \mathbf{y}_N)$ to denote all the variables of the optimization. Thus, the objective becomes the linear function $q - s$ with an additional constraint that

$$\sqrt{\rho \sum_i l_i^2} \leq s \tag{11}$$

The above is same as $||A\mathbf{r}||_2 \leq \mathbf{c}^T \mathbf{r}$ for the constant matrix $A$ (with entries 0 or $\sqrt{\rho}$) and constant vector $\mathbf{c}$ (with 1 in the $s$ component, rest 0's) that picks the $l_i$'s and $s$ respectively. This is a SOCP form of constraint, and the linear objective $q - s$ makes the problem after this transformation a MISOCP. In the above reformulation, the only approximation is introduced in writing $F_i$ as a linear fractional term. One way of such approximation is via discretization. Suppose the fraction function $F(\mathbf{z}, \widehat{\mathbf{x}}_i)$ has a separable (in $\mathbf{z}$) numerator and denominator of the form $\frac{\sum_j n(z_j, \widehat{\mathbf{x}}_i)}{\sum_j d(z_j, \widehat{\mathbf{x}}_i)}$ where $j$ ranges over the components of $\mathbf{z}$, and $n(z_j, \widehat{\mathbf{x}}_i)$ and $d(z_j, \widehat{\mathbf{x}}_i)$ are non-negative and Lipschitz continuous in $z_j$ with Lipschitz constants $C^n, C^d$ respectively. In this case, a general approximation via discretization is possible with the following guarantee:

**Theorem 2.** *For $F(\mathbf{z}, \widehat{\mathbf{x}}_i) = \frac{\sum_j n(z_j, \widehat{\mathbf{x}}_i)}{\sum_j d(z_j, \widehat{\mathbf{x}}_i)}$ as stated above and approximated as $\frac{\mathbf{a}_i^T \mathbf{v} + b_i}{\mathbf{a}_i'^T \mathbf{v} + b_i'}$, an approximation via discretization of $z_j$ with $K$ pieces yields $|\mathcal{G}(\mathbf{z}^*) - \mathcal{G}(\widetilde{\mathbf{z}}^{**})| \leq O(\max\{C^n, C^d\}/K)$, where $\mathcal{G}(\mathbf{z}^*)$ and $\mathcal{G}(\widetilde{\mathbf{z}}^{**})$ are the optimal objective values with approximation (MISOCP) and without the approximation respectively.*

Next, we show instantiation of the just presented general recipe for three widely studied problems.

### 3.2 Applications

**Notation**: In the SSG (facility location) application $m$ resources (facility) are allocated to $M$ targets (locations). $\mathbf{x}$ maps to type $\theta_{\mathbf{x}}^a$ of adversary, and type $\theta_{\mathbf{x}}^d$ of defender in SSG, and type $\theta_{\mathbf{x}}$ of clients of facility or directly $V_{\mathbf{x}}$ utility for each client type in facility location.

**Bayesian Stackelberg Security Game with Quantal Response**: A SSG models a Stackelberg game where a defender moves first to allocate $m$ security resources for protecting $M$ targets. The randomized allocation is specified by decision variables $\mathbf{z}$ of dimension $M$ with the constraints that $\sum_{i=1}^{M} z_i \leq m$ ($z_i \in [0, 1]$); $z_i$ is interpreted as the protection probability of the target $i$. Past works have used the model of a quantal responding adversary [Sinha et al., 2018]]. We generalize this to a Bayesian game version where there is a continuum of attackers types with the type specified by a parameter $\mathbf{x}$ and an *unknown* prior distribution over these types. The attacker's utility in attacking the target $j$ is a function of the protection probability of target $j$ and type: $h(z_j, \theta_{\mathbf{x}}^a)$. Similarly, the defender's utility when target $j$ is attacked is: $u(z_j, \theta_{\mathbf{x}}^d)$ for some player-specific parameters $\theta$ that depend on $\mathbf{x}$. Following quantal response model (for attacker only), the attacker of type $\mathbf{x}$ attacks a target $j$ with probability $\frac{\exp(h(z_j, \theta_{\mathbf{x}}^a))}{\sum_{j \in [M]} \exp(h(x_j, \theta_{\mathbf{x}}^a))}$ and the defender utility is $F(\mathbf{z}, \mathbf{x}) = \frac{\sum_{j \in [M]} u(z_j, \theta_{\mathbf{x}}^d) \exp(h(z_j, \theta_{\mathbf{x}}^a))}{\sum_{j \in [M]} \exp(h(x_j, \theta_{\mathbf{x}}^a))}$. Note that in case the defender's utilities $u(z_j, \theta_{\mathbf{x}}^d)$ take negative values and the assumptions of Theorem 2 will be violated. This issue can be simply fixed by choosing $\alpha$ such that $\alpha \geq \max_{\mathbf{z}, \mathbf{x}} \{-u(z_j, \theta_{\mathbf{x}}^d)\}$ and replacing $F(\mathbf{z}, \mathbf{x})$ by $F(\mathbf{z}, \mathbf{x}) + \alpha = \frac{\sum_{j \in [M]} (u(z_j, \theta_{\mathbf{x}}^d) + \alpha) \exp(h(z_j, \theta_{\mathbf{x}}^a))}{\sum_{j \in [M]} \exp(h(x_j, \theta_{\mathbf{x}}^a))}$. This will make all the numerators and enumerators of the objective function non-negative, while keeping the same optimization problem. We also note that quantal response is also known as multinomial logit model in the discrete choice model literature [Train, 2003]. Our generalization here to multiple types of adversary makes the problem akin to the mixed logit model in discrete choice models, which is generally considered intractable. As a consequence, our solution addresses a basic problem in discrete choice models also.

Following our set-up, we observe $N$ samples of the types of attackers $\widehat{\mathbf{x}}_1, \ldots, \widehat{\mathbf{x}}_N$ (which gives $\widehat{\theta}_1^a, \ldots, \widehat{\theta}_N^a, \widehat{\theta}_1^d, \ldots, \widehat{\theta}_N^d$) and we solve a robust version of the problem. Further, following our general recipe for solving the robust problem, we piecewise approximate the numerator and denominator of $F$ using $K$ pieces, where the dimension of $\mathbf{v}$ is $d = MK$. For this approximation, we require two

additional linear constraints over the constraints in Equations (2-35). The optimization then is:

$$\max_{\mathbf{r}} \; q - s \qquad\qquad\qquad\qquad\qquad\qquad \text{(SSG)}$$

$$\text{subject to Constraints (2-35)}, \sum_{j \in [M]} \sum_{k \in [K]} v_{jk} - mK \le 0, v_{j,k} \ge v_{j,k+1}; \quad \forall k \in [K]$$

The overall additive solution bound of $O(1/K)$ can be readily inferred from Theorem 2.

**Max-Capture Competitive Facility Location (MC-FLP):** In this problem [Mai and Lodi, 2020], a firm has $M$ locations ($[M]$) to set up at most $m < M$ facilities. The aim is to maximize the number of clients using this firm's facilities. The competitor(s) already have facilities running at locations $Y \subset [M]$. There are different types of clients, where types are denoted by $\mathbf{x}$. The number of clients of type $\mathbf{x}$ is known and equal to $s_{\mathbf{x}}$. However, the distribution over types is *unknown*. The firm's decision of which location to choose is given by binary variables $z_j \in \{0, 1\}$ for $j \in [M]$. A utility of any client of type $\mathbf{x}$ for visiting location $j$ is $V_{\mathbf{x},j}$. The *choice probability* of a client of type $\mathbf{x}$ choosing any of this firm's location is given as a quantal response model $\frac{\sum_{j \in [M]} z_j e^{V_{\mathbf{x},j}}}{\sum_{j \in [M]} z_j e^{V_{\mathbf{x},j}} + \sum_{j \in Y} e^{V_{\mathbf{x},j}}}$.
For shorthand, we abuse notation and use $V_{\mathbf{x},j}$ to replace $e^{V_{\mathbf{x},j}}$ and $U_{\mathbf{x},Y}$ to replace $\sum_{j \in Y} e^{V_{\mathbf{x},j}}$. This gives $F(\mathbf{z}, \mathbf{x}) = \frac{s_{\mathbf{x}} \sum_{j \in [M]} z_j V_{\mathbf{x},j}}{\sum_{j \in [M]} z_j V_{\mathbf{x},j} + U_{\mathbf{x},Y}}$, which is interpreted as the expected number of clients of type $\mathbf{x}$ choosing this firm's facilities.

Following our set-up, we observe $N$ samples of the types of clients samples $(\widehat{V}_{1,j})_{j \in [M]}, \ldots, (\widehat{V}_{N,j})_{j \in [M]}$ and we solve a robust version of the problem. Here, we get $F_i = \frac{s_i \sum_{j \in [M]} z_j \widehat{V}_{i,j}}{\sum_{j \in [M]} z_j \widehat{V}_{i,j} + \widehat{U}_{i,Y}}$. Next, following our general recipe for solving the robust problem, we note that $F_i$ is already in the form $\frac{\mathbf{a}_i^T \mathbf{v} + b_i}{\mathbf{a}_i'^T \mathbf{v} + b_i'}$ where $\mathbf{z}$ plays the role of $\mathbf{v}$. Thus, the dimension of $\mathbf{v}$ is $M$ and no approximation is needed here for $F_i$; by Theorem 2, we achieve the global optimal solution by solving MISOCP optimally. The full MISOCP with an additional number of location constraint is:

$$\max_{\mathbf{r}} q - s \text{ subject to Constraints (2-35)}, \sum_{j \in [M]} v_j - m \le 0.$$

**Max-Capture Facility Cost Optimization (MC-FCP):** In the previous **MC-FLP** problem, the budget was specified as a constraint on the number of facilities. However, often a more realistic set-up is where there is a monetary constraint and the attractiveness of a facility depends on the investment into the facility. Thus, modifying the previous problem slightly, $z_j$ takes a different meaning of the amount of investment into facility at location $j$ (zero investment indicates no facility). Given this investment, the attractiveness of a facility $j$ for the client of type $\mathbf{x}$ is given as $h(z_j, \theta_{\mathbf{x},j})$ for some parameter $\theta$ dependent on $\mathbf{x}$ and $j$. And the *choice probability* of a client of type $\mathbf{x}$ choosing any of this firm's location is given as a quantal response model $\frac{\sum_{j \in [M]} e^{h(z_j, \theta_{\mathbf{x},j})}}{\sum_{j \in [M]} e^{h(z_j, \theta_{\mathbf{x},j})} + U_{\mathbf{x},Y}}$. This gives $F(\mathbf{z}, \mathbf{x}) = \frac{s_{\mathbf{x}} \sum_{j \in [M]} e^{h(z_j, \theta_{\mathbf{x},j})}}{\sum_{j \in [M]} e^{h(z_j, \theta_{\mathbf{x},j})} + U_{\mathbf{x},Y}}$, which is interpreted similar to **MC-FLP**. As stated, we observe $N$ samples of the types of clients which gives $(\widehat{\theta}_{1,j})_{j \in [M]}, \ldots, (\widehat{\theta}_{N,j})_{j \in [M]}$ and we solve a robust version of the problem. Here, we get $F_i = \frac{s_i \sum_{j \in [M]} e^{h(z_j, \widehat{\theta}_{i,j})}}{\sum_{j \in [M]} e^{h(z_j, \widehat{\theta}_{i,j})} + \widehat{U}_{i,Y}}$. Next, as can be seen from the form, this is similar to the **SSG** problem and, following our general recipe, with two additional linear constraints the optimization formulation is exactly same as Equation (SSG).

## 4   Scaling up in Number of Samples

The transformation to a MISOCP helps in scalability over a general mixed integer concave program, but for real world dataset sizes (e.g., $80,000$ data points in our experiments) we need further scalability. We explore two related techniques towards this end: clustering and stratified sampling. For both approaches, we obtain a representative subset of $S$ data points ($S \ll N$) and a modified weighted objective, which converts to a much smaller tractable MISOCP compared to the original problem.

For solution guarantees, we need mild assumptions: in particular, for the rest of this section we assume a bounded $F$, i.e., for some fixed $\psi$ $\max_{\mathbf{z}}\{F(\mathbf{z}, \mathbf{x})\} - \min_{\mathbf{z}}\{F(\mathbf{z}, \mathbf{x})\} \leq \psi^2$ $\forall \widehat{\mathbf{x}}_1, \ldots, \widehat{\mathbf{x}}_N$ and $\tau$-lipschitzness of $F$ in the argument $\mathbf{x}$: $|F(\mathbf{z}, \mathbf{x}') - F(\mathbf{z}, \mathbf{x})| \leq \tau ||\mathbf{x}' - \mathbf{x}||_2$ $\forall \mathbf{z}$.

**Clustering Approach**: We cluster the $N$ points $\mathbf{x}_1, \ldots \mathbf{x}_N$ into $S$ groups and for each group $s$ we have $||\mathbf{x}_i - \mathbf{x}^s|| \leq \epsilon$, where $\mathbf{x}^s$ is the cluster center of cluster $s$. We call $\epsilon$ the clustering radius. Let $C_s$ be the number of points in the cluster $s$, hence $\sum_{s \in [S]} C_s = N$. We use a shorthand for the original objective function of the MISOCP $\mathcal{G}(\mathbf{z})$:

$$\sum_i \frac{F(\mathbf{z}, \widehat{\mathbf{x}}_i)}{N} - \sqrt{\rho \sum_i \Big( \sum_i \frac{F(\mathbf{z}, \widehat{\mathbf{x}}_i)}{N} - F(\mathbf{z}, \widehat{\mathbf{x}}_i) \Big)^2} = \widehat{\mathrm{Mean}}(F(\mathbf{z}, \mathbf{x})) - \sqrt{\rho \widehat{\mathrm{Var}}(F(\mathbf{z}, \mathbf{x}))}$$

where $\widehat{\mathrm{Mean}}$ is empirical mean and $\widehat{\mathrm{Var}}$ is *unnormalized variance*. After clustering, we solve for the same problem but only with cluster centers and appropriate *weighing*, to get modified objective $\widehat{\mathcal{G}}(\mathbf{z})$:

$$\sum_s C_s \frac{F(\mathbf{z}, \mathbf{x}^s)}{N} - \sqrt{\rho \sum_s C_s \Big( \sum_s C_s \frac{F(\mathbf{z}, \mathbf{x}^s)}{N} - F(\mathbf{z}, \mathbf{x}^s) \Big)^2} = \widehat{\mathrm{Mean}}^S(F(\mathbf{z}, \mathbf{x})) - \sqrt{\rho \widehat{\mathrm{Var}}^S(F(\mathbf{z}, \mathbf{x}))}$$

The conversion to MISOCP is exactly the same, except for $F_i$'s being weighted as shown above; details of conversion are in the appendix. We bound the approximation incurred by the two terms above (weighted mean and unnormalized weighted variance) separately below

**Lemma 1.** *Under assumptions stated above, we have* $\left|\widehat{Mean}(F(z, x)) - \widehat{Mean}^S(F(z, x))\right| \leq \tau\epsilon$ *and* $\left|\sqrt{\rho\widehat{Var}(F(z, x))} - \sqrt{\rho\widehat{Var}^S(F(z, x))}\right| \leq (\psi + \sqrt{2\tau\epsilon})\sqrt{\frac{2\tau\epsilon\xi}{N}}.$

The next result is obtained by using the lemma above

**Theorem 3.** *Given $\widehat{z}$ an optimal solution for $\max_z \widehat{\mathcal{G}}(z)$ (clustering approximation) and $z^*$ optimal for MISOCP $\max_z \mathcal{G}(z)$, the following holds: $|\mathcal{G}(\widehat{z}) - \mathcal{G}(z^*)| \leq 2(\tau\epsilon + \psi\sqrt{\frac{2\tau\epsilon\xi}{N}} + \frac{2\tau\epsilon\xi}{\sqrt{N}}).$*

**Stratified Sampling**: Similar in spirit to clustering, the space of $\mathbf{x}$ space is divided into $T$ strata. Each strata has $C_t$ samples, such that $\sum_{t \in [T]} C_t = N$. Next, distinct from the clustering approach, we draw $N_t$ samples randomly from the $t^{th}$ stratum with a total of $\sum_t N_t = S$ samples (note same number of total samples $S$ as in clustering). For each stratum $t$ we have $||\mathbf{x}_i - \mathbf{x}_j|| \leq d_t$ for any $\mathbf{x}_i, \mathbf{x}_j$ in stratum $t$. We denote a random sample in stratum $t$ as $\widehat{\mathbf{x}}^j$ where $j \in [N_t]$ (note superscript is to distinguish from the subscript used to index all the $\widehat{\mathbf{x}}$'s). This approach is the preferred one if the clustering approach results in cluster centers that are not allowed as parameter values (e.g., cluster center may be fractional where $\mathbf{x}$'s can only be integral).

Let $l_t = \frac{C_t}{N_t}$. Use $\widehat{Mean}^T(F(\mathbf{z}, \mathbf{x}))$ to stand for $\frac{1}{N} \sum_{t \in [T]} l_t \sum_{j \in [N_t]} F(\mathbf{z}, \widehat{\mathbf{x}}^j)$ and $\widehat{Var}^T(\mathbf{z}, \widehat{\mathbf{x}})$ for $\sum_{t \in [T]} l_t \sum_{j \in [N_t]} \left( \widehat{Mean}(F(\mathbf{z}, \mathbf{x})) - F(\mathbf{z}, \widehat{\mathbf{x}}^j) \right)$. After stratified sampling our modified weighted objective $\widehat{G}(\mathbf{z})$ is $\widehat{Mean}^T(F(\mathbf{z}, \mathbf{x})) - \sqrt{\rho\widehat{Var}^T(\mathbf{z}, \widehat{\mathbf{x}})}$. Next, similar to clustering, bounds for $\widehat{Mean}^T$ and $\widehat{Var}^T$ but with high probability (Lemma 2,3 in appendix) lead to the main result:

**Theorem 4.** *Let $D = \max_{z, x} |F(z, x)|$ for bounded function $F$. Under mild assumptions, and $\widehat{z}$ an optimal solution for $\max_z \widehat{\mathcal{G}}(z)$ and $z^*$ optimal for MISOCP $\max_z \mathcal{G}(z)$, and $N_* = \min_t N_t$, the following statement holds with probability $\geq 1 - 2\sum_t \exp^{\frac{-2\sqrt{N_*}\epsilon^2}{\tau^2 d_t^2}} - 4\sum_t \exp^{\frac{-2\sqrt{N_*}\epsilon^2}{4\tau^2 d_t^2 D^2}}$ :*

$$|\mathcal{G}(\widehat{z}) - \mathcal{G}(z^*)| \leq \frac{2\epsilon}{(N_*)^{1/4}} \left( 1 + 2\sqrt{\frac{\xi}{\widehat{Var}(F(z, x))}} \right).$$

Thus, with increasing samples in all strata, optimality gap approaches 0 with prob. approaching 1.

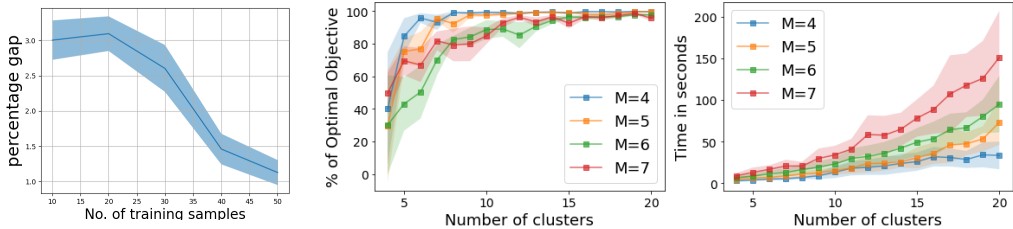

Figure 1: (Left) % gap between the utility of decisions using true and learned regressor with varying training data size $N_T$. (Middle) Objective value achieved using clustering approach as a % of **OPT**. (Right) Time to solve each optimization to optimality. Middle and right results are shown for varying alternatives number $M$. Underlying parameters are $N = 500, m = 1, \xi =$1E6.

Table 1: We use clustering/stratified sampling to approximately solve a problem (**Approx-OPT**). The table shows the mean percentage: $\frac{100 \times \textbf{Approx-OPT}}{\textbf{OPT}}$ and standard deviation over 10 different synthetic SSG datasets with underlying parameters $N = 500, M = 6, m = 1, K = 10$.

| Method | Total no. of samples (S) | | |
|---|---|---|---|
| | 8 | 16 | 24 |
| Clustering | $80.17 \pm 2.14$ | $90.74 \pm 1.20$ | $94.03 \pm 0.57$ |
| 1 per strata | $85.49 \pm 1.94$ | $94.59 \pm 0.76$ | $99.25 \pm 0.39$ |
| 2 per strata | $91.78 \pm 1.33$ | $94.55 \pm 0.64$ | $99.54 \pm 0.25$ |
| 4 per strata | $78.89 \pm 1.73$ | $94.86 \pm 0.94$ | $98.77 \pm 0.39$ |
| 8 per strata | $92.19 \pm 0.67$ | $95.85 \pm 0.80$ | $99.37 \pm 0.25$ |
| Uniform sampling (no cluster/strata) | $73.92 \pm 33.71$ | $78.08 \pm 28.62$ | $79.86 \pm 26.81$ |

Table 2: Objective values as a % of **OPT** across various methods repeated over 5 synthetic SSG datasets with parameters $N = 500, M = 10, m = 1$ for varying regularization ($\xi$, on left) and $N = 500, m = 1, \xi = $1E6 for varying no. of targets (M, on right). The no. of clusters/strata is 50.

| Method | Regularization ($\xi$) | | | | No. of Alternatives (M) | | |
|---|---|---|---|---|---|---|---|
| | 1E3 | 1E4 | 1E5 | 1E6 | 10 | 25 | 50 |
| TT-GAD | $99.8\pm0.1$ | $99.4\pm0.1$ | $92.7\pm0.3$ | $82.6\pm0.4$ | $82.6 \pm 0.4$ | $90.2 \pm 0.5$ | $92.2 \pm 0.3$ |
| PGA | $98.9\pm0.1$ | $98.1\pm0.2$ | $87.7\pm0.5$ | $49.2\pm0.9$ | $49.2 \pm 0.9$ | $90.9 \pm 0.7$ | $93.5 \pm 0.4$ |
| Clustering | $99.9\pm0.1$ | $99.9\pm 0.1$ | $99.8\pm0.1$ | $99.6\pm0.1$ | $99.6 \pm 0.1$ | $99.5 \pm 0.2$ | $99.4 \pm 0.1$ |
| Sampling | $100.0\pm0.0$ | $99.9\pm 0.1$ | $99.9\pm0.1$ | $99.8\pm0.1$ | $99.8 \pm 0.1$ | $99.6 \pm 0.1$ | $99.5 \pm 0.2$ |

## 5 Experiments

We evaluate our methods on (a) Stackleberg Security Games (**SSG**) with Quantal Response (synthetic data), (b) Maximum capture Facility Location Planning (**MC-FLP**) and (c) Maximum capture Facility Cost Planning (**MC-FCP**). Empirically we demonstrate (i) better solution quality of our method compared to baselines, (ii) practical scalability of our method, and (iii) improvement over non-robust optimization on those data points that contribute least to the objective (akin to rare classes in classification) while not sacrificing average performance. We fix $K = 10$ in approximation via discretization as we find that objective increase saturates for this $K$ (see Appendix K). We use a 2.1 GHz CPU with 128GB RAM.

**Baselines:** We use the following two methods as baselines: (i) Projected Gradient Ascent (**PGA**) on the formulation (VR), (ii) Two Time Scale Gradient Ascent Descent (**TT-GAD**) [Lin et al., 2020] on the formulation (DRO) where the inner minimization is convex and the outer maximization is non-concave. The numbers reported for our baselines are the *best values* over 10 random initializations.

### 5.1 SSG with Quantal Response (Synthetic Data)

We generate attacker and defender utilities following Yang et al. [2012]; a complete description of data generation and choice of $f^*$ is in Appendix I. We generate five datasets of size $N = 500$ each in order to observe the variance of every result reported in this sub-section; this is also the largest size

Table 3: Average client choice probabilities for availing the facility across various settings. H denotes average over those 5% of the clients in test data with the lowest choice probabilities, A denotes average over all the samples in the test set.

| | MC-FCP | | | | | | MC-FLP | | | | | |
|---|---|---|---|---|---|---|---|---|---|---|---|---|
| $\xi$ | m=7 | | m=10 | | m=13 | | m=10 | | m=12 | | m=14 | |
| | H | A | H | A | H | A | H | A | H | A | H | A |
| ERM | 0.069 | 0.692 | 0.150 | 0.719 | 0.420 | **0.758** | 0.175 | 0.741 | 0.426 | **0.769** | 0.469 | 0.772 |
| 1E2 | 0.069 | 0.692 | **0.417** | **0.751** | 0.4170 | 0.751 | **0.418** | **0.763** | 0.426 | **0.769** | 0.469 | 0.772 |
| 1E3 | **0.093** | **0.697** | 0.416 | 0.747 | 0.417 | 0.757 | **0.418** | **0.763** | 0.425 | 0.768 | 0.533 | **0.777** |
| 1E4 | **0.093** | **0.697** | 0.416 | 0.747 | **0.446** | 0.750 | 0.417 | 0.759 | **0.531** | 0.767 | **0.539** | 0.769 |

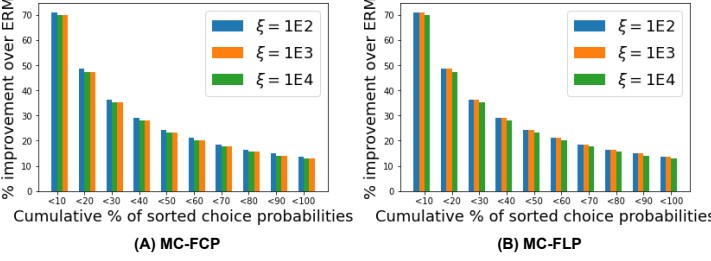

Figure 2: The bar plots show the percentage improvement of choice probabilities of our clustering approach over ERM over cumulative buckets of choice probabilities in ascending order with varying regularization $\xi$ with fixed budget $m$=10. The buckets are made by sorting all the clients in the test set by ascending choice probabilities and then considering cumulative buckets as the first 5%, the first 10% and so on.

that we could solve exactly optimally within an hour using all the data points. First, we empirically validate Theorem 1 by plotting in Figure 1(left) the relative gap between the true utility $E_{P^*}[F(\cdot, \mathbf{x})]$ of the decisions output by running DRO on the output $(\mathbf{x}^*)_{i \in [N]}$ of the true $f^*$ (assumed fixed linear function) versus on $(\widehat{\mathbf{x}})_{i \in [N]}$ from learned $f$, as the training data size $N_T$ for learning $f$ is varied.

Next, we focus on only using $(\widehat{\mathbf{x}})_{i \in [N]}$ and the optimal solution for $(\widehat{\mathbf{x}})_{i \in [N]}$ is named as **OPT**. Figure 1 (middle) demonstrates empirically that the solution of the optimization problem on cluster centers converges to **OPT** with only a few number of clusters and the time for the optimization shown in Figure 1 (right) is reasonable. Next, the results in Table 1 show a comparison of the clustering and stratified sampling approach using the metric of how close they get to **OPT**. We find stratified sampling to be better than clustering in almost all cases and a simple uniform sampling (no cluster/strata) fails to return a solution close to **OPT**. Table 2 (left) demonstrates that the baselines struggle to reach the optimal value objective as the magnitude of regularization ($\xi$) increases. Intuitively, as $\xi$ increases the variance term (which is highly non-convex) contributes more to the objective and stationary points reached by the baselines are quite sub-optimal compared to the global optimal. In addition, with increasing $\xi$ the ambiguity set becomes larger possibly containing more local optimal solutions. We also study varying the parameter $M$ ($m$ fixed) and Table 2 (right) shows that our approaches outperform the gradient-based baselines across different values of $M$.

## 5.2 MC-FLP and MC-FCP (Real Data)

**P&R-NYC Dataset :** We use a large and challenging Park-and-ride (P&R) dataset collected in New York City, which provides utilities for 82341 clients ($N$) for 59 park and ride locations ($M$), along with their incumbent utilities for competing facilities [Holguin-Veras et al., 2012]; this data was directly used for **MC-FLP**. For **MC-FCP** we additionally use generated costs, which are not present in the P&R data. A complete description of data generation is in Appendix I. Both these problems could not be solved at all with our MISOCP alone (no clustering) as the optimization did not finish in 24 hours. Hence, we use our clustering approach with 50 clusters.

We compare to a baseline solution of the non-robust empirical risk minimization (ERM) (also called the sample average approximation or SAA). We split the data (randomly) into training and test (80:20) and then obtain the decision $\widehat{\mathbf{z}}$ using the training data. Then, we obtain the choice probability (recall this as probability of a client choosing any of the firm's facility) for every client in the test data for the

decision $\widehat{\mathbf{z}}$. We compare the performance of ERM and our method for clients (in test set) bucketed by choice probabilities in Figure 2 and Table 3. The buckets are made by sorting all the clients in the test set by ascending choice probabilities and then considering cumulative buckets as the first 5%, the first 10% and so on. In Fig. 2, we show that the average percentage improvement in choice probabilities of our robust approach over ERM is considerably higher for clients with lower choice probability (these clients contribute least to the objective) and the over all average over all clients (rightmost on x-axis) is slightly better than ERM. In Table 3, note the significantly increased probabilities for low choice probability clients (low choice prob. using $\widehat{\mathbf{z}}$) without compromising the average performance across all clients for varying $M$. Additional results are in Appendix J.

## 6  Conclusion

We presented an approach for a distributionally robust solution to a class of non-convex sum of fractional solutions, with guaranteed near global optimality. We presented application to three prominent practical problems and the connection to discrete choice models opens up possibilities of applying our approach to even more problems. Further investigation on how to cluster or stratify more effectively (than k-means) to achieve even more scalability is a possible future research direction. Further, we used a $\chi^2$-divergence based ambiguity set, which only covers nearby distributions with same support as the given data samples; exploring Wasserstein ambiguity sets is a possible future research direction. We hope that our work inspires tackling robust formulation of more classes of non-convex problems, with guarantees for global optimality.

## Acknowledgement

This research/project is supported by the National Research Foundation, Singapore under its AI Singapore Programme (AISG Award No: AISG2-RP-2020-017).

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
