# OpenReview forum: "Scalable Distributional Robustness in a Class of Non-Convex Optimization with Guarantees"
_NeurIPS.cc/2022/Conference — NeurIPS 2022 Accept_

### Official Review · Reviewer_bkf3 · 2022-07-10

**Rating:** 7
**Confidence:** 1
**Soundness:** 4 excellent
**Presentation:** 3 good
**Contribution:** 3 good

**Summary:**

The paper considers optimization problems for decision variables with a robust requirement to the observed data.
Thus the authors consider distributionally robust optimization and recast it via the variance regularized form into a mixed-integer second-order cone program (MISOCP). This formulation allows to efficiently obtain a solution while retaining the guarantees of the original formulation.
To further improve scalability the authors propose two approaches: i) clustering and ii) stratified subsampling. Both approaches can be shown to be close to the original solution (depending on the number of clusters or the samples taken).

As applications, the paper presents 3 game theoretic optimization tasks (a Bayesian version of Stackelberg Security games and two versions of max-capture facility assignment) that can be phrased to match this setting.
Experimentally, the paper shows that i) the proposed algorithm outperforms several baselines and that the clustering and sampling approaches perform close to the original problem while only utilizing s small amount of data.



**Questions:**

Based on the above:
- Can you motivate the setting: SSG-style decision problems + DRO?
- Can you compare to a non-DRO (and non-VR) baseline to show the benefit of the distributional approach empirically?
- Can you motivate the need for speed-up/scalability for the one-time optimization cost?
- Wouldn't subsampling, e.g. at uniformly, $S$ of $N$ samples (without stratification) be another baseline? How does stratification improve upon this?



**Limitations:**

Technological limitations seem sufficiently contextualized.
The paper does not discuss the negative societal impact of the work, but I don't think particular consideration for this is needed.



**Strengths And Weaknesses:**

This paper is outside my area of expertise, which I reflected by a low confidence score.

Strengths:
- Overall well written. Small improvements are possible (see "Presentation" below).
- Approach seems mathematically sound.
- Clustering and stratified sampling seem surprisingly simple yet elegant solutions for speed-ups.

Weaknesses:
- Evaluation on only 2 datasets (one synthetic, one real).
- Some motivation is unclear (see below).

Presentation:
I found the applications section hard to follow, some small examples or illustrations might benefit the reader.

Motivation & Evaluation:
While I know (though not well) that SSGs are important applications in some areas and DRO may be important settings, I think the paper did not motivate well why SSG-style decision problems need DRO on both a conceptual and an empirical level. All baselines use some version of DRO or variance reduction. It would be interesting to see how these compare to non-DRO approaches to justify why the setting is important.

Further, can you clarify the need for the optimizations (clustering, stratified sampling) you discuss? The cost of solving the unoptimized MISCOP is a one-time cost.
Even in the setting in 5.2 one could either ju1st spend the one time for optimization once or else just subsample $S$ samples from $N$ (in a non-stratified way).

---

> ### Author Response · Authors · 2022-08-02
> **Thank you for the Review**
>
> We thank the reviewer for the comments.
> We add some discussion below in Appendix J and illustrative example in new section Appendix M (given the limited space in the main paper). We do add the random sampling alternative result in the main paper in Table 1.
>
> ***Why DRO for SSG?*** The SSG problem with quantal response has been used in wildlife protection [1]. A simplifying assumption of one type of attacker (with learnable parameter from attack data) has been used in such SSG problems, but realistically attackers are not all same and there is one rare attempt to account of different types of attackers [2], which forces a Gaussian prior on attacker types. Indeed, this prior should be estimated from data, but there is limited data [2] for the problem, thus, being robust to the empirical prior distribution revealed by data is definitely helpful. Our DRO results show substantial improvement over non-DRO approaches (see the paragraph below), further supporting why DRO approach can be useful for SSG.
>
> ***Baselines***: The reviewer points out that all the baselines on the experiments are on some form of DRO, but as mentioned in line 319 and in Table 3, we appropriately compare DRO solutions with ERM (the non-DRO problem) solutions. Further, Figure 2 shows the improvement over the baseline ERM, supporting the need for DRO over non-DRO approaches.
>
> ***Need for speed up and one time cost***: The problem at hand scales exponentially both in memory and time, so solving on real world datasets such as the Max Capture Facility dataset of 80,000 datapoints is simply infeasible on regular computers as the program does not even load on a machine with 128GB RAM. It is known that for SSG decisions change monthly as new attack data is received [1] and the tool runs on resource constrained computers. Similarly, facility cost optimization decisions can also change with changing profile of customers and/or change in type of services or promotions offered (revealed in newly collected data). Thus, the DRO optimization can run repeatedly at given frequencies and needs to be efficient in practice.
>
> ***Random sampling***: We agree that random sampling can act as a baseline, but random samples would have a high variance when estimating the mean and variance of all data (requiring more samples than cluster centers or stratified samples to get lower variance). We conducted experiments using random sampling and the results are stated below (as well as added to Table 1 in the main paper):
>
> We use clustering/stratified sampling to approximately solve a problem (***Approx-OPT***). The table shows the mean percentage: $\frac{100 \times \textbf{Approx-OPT}}{\textbf{OPT}}$ and  standard deviation over 10 different synthetic SSG datasets with underlying parameters $N=500, M=6, m=1, K=10$. For each dataset, the best value out of 10 random samples was selected.
> || | | | |
> |--|:--:|:--:|:--:|:--:|
> | Method | | Total number of samples (S) | |
> | | 8 | 16 | 24 |
> |Clustering | 80.17 $\pm$ 2.14 | 90.74 $\pm$ 1.20 | 94.03 $\pm$ 0.57||
> |1 per strata | 85.49 $\pm$ 1.94 | 94.59 $\pm$ 0.76 | 99.25 $\pm$ 0.39||
> |2 per strata | 91.78 $\pm$ 1.33 | 94.55 $\pm$ 0.64 | 99.54 $\pm$ 0.25||
> |4 per strata | 78.89 $\pm$ 1.73 | 94.86 $\pm$ 0.94 | 98.77 $\pm$ 0.39||
> |8 per strata | 92.19 $\pm$ 0.67 | 95.85 $\pm$ 0.80 | 99.37 $\pm$ 0.25||
> |Uniform | 73.92 $\pm$ 33.71 | 78.08 $\pm$ 28.62 | 79.86 $\pm$ 26.81||
>
> [1] Fang, Fei, et al. "Deploying PAWS: Field Optimization of the Protection Assistant for Wildlife Security." AAAI. Vol. 16. 2016.
>
> [2] Yang, Rong, et al. "Adaptive resource allocation for wildlife protection against illegal poachers." Aamas. 2014.

---

> > ### Comment · Reviewer_bkf3 · 2022-08-07
> > **Thank you!**
> >
> > Thank you for your response.
> > As an outsider to the field I found the additional explanations and baseline quite helpful in contextualizing the work.
> > I have raised my contribution and overall score to reflect this.
> >
> > While I feel positive about the paper (in terms of technical soundness and contribution), this still is still very much outside my field and I may have missed crucial details.
> >
> > Best,
> > Reviwer bkf3

---

> > > ### Author Response · Authors · 2022-08-07
> > > **Thanks a lot!**
> > >
> > > We thank the reviewer for the kind words and for raising the scores.

---

### Official Review · Reviewer_tnSW · 2022-07-11

**Rating:** 7
**Confidence:** 2
**Soundness:** 4 excellent
**Presentation:** 3 good
**Contribution:** 3 good

**Summary:**

This paper optimizes the non concave fractional objectives with variance regularizer, and reformulate it as a mixed integer concave program with all constraints linear. The key observation is when the individual objective $F_i$ can be written approximately as a linear fraction with affine numerator and denominator, the non-convex constraints can be converted into convex constraints by applying the McCormick technique. The MICP can be further converted into a MISOCP for better scalability. Such approximation of the fractional objective is possible via PWLA with sub-linear approximation rate. To further relax the number of constraints in solving MISOCP, this paper consider two more techniques. The first technique is the clustering approach, where cluster centers are used as new data points, the approximation error depends sub-linearly on the clustering radius and the number of original data points. The second technique is stratified sampling, which is similar as the cluster approach but real data points are used rather than the cluster centers. Three application problems are given to conduct experiments. The SSG with quantal response application is conducted on small synthetic datasets. Both clustering approach and stratified sampling approach outperform baseline algorithms. Both MC-FLP and MC-FCP applications uses real data, and the clustering approach is considered. Their method outperforms ERM in most of the cases.

**Questions:**

1. I have seen the idea of convex relaxations elsewhere in learning, namely, in robust optimization and adversarial learning. Can the author comment on the origin of this idea?

Wong and Kolter. Provable defenses against adversarial examples via the convex outer adversarial polytope. ICML 2018.
Raghunathan et al. Certified Defenses against Adversarial Examples. ICLR 2018.

2. For the experiment, it is not clear to me why PGA is chosen for VR and TT-GAD is chosen for DRO, not vice versa. A follow up question,  why these two algorithms are chosen? As they seem to be preliminary methods in the first-order optimization algorithm family, there are other algorithms tailored for DRO.

Qi et al. An Online Method for A Class of Distributionally Robust Optimization with Non-Convex Objectives. NeurIPS 2021.

**Limitations:**

1. Theorem 1 for learning with DRO directly should be compared in depth with the bound obtained by the proposed MISOCP if possible. For example, does Theorem 1 shed light on how to choose the number of piecewise linear functions, the number of cluster centers and the cluster radius?
2. While the experiments focuses the performance of the proposed MISOCP, I think the computational time should also be consider, especially the time consumed during the clustering process, then compare the total computational time with baselines.

**Strengths And Weaknesses:**

Strengths:
1. Clever idea and solid analysis. For the idea part, the main contribution is to convert the original non-concave problems into a mixed integer concave program, therefore the near global optimality can be guaranteed. Furthermore, as programming approach is usually less favored due to the scalability, this paper proposes two relaxation techniques and seems effective in the experiments. For the analysis, every transformation of the problem comes with analysis on the approximation error, and dependence on the hyperparameters are clearly presented.
2. Well written paper for experienced readers. I am not very familiar with mix integer concave program but I find the paper is easy to follow, as the motivation and technical tools are well explained.

Weaknesses:
The paper provides a whole package on this particular optimization problem and I do not see obvious weaknesses.

---

> ### Author Response · Authors · 2022-08-02
> **Thank you for the Review**
>
> We thank the reviewer for the comments and pointing out that we provide a whole package on this problem.
>
> ***Convexification***: Indeed, as the reviewer points out, convexification is a standard approach to handle non-convex optimization problems. However, different non-convex functions require different techniques for obtained a tight convex relaxation. Raghunathan et al [1], relax optimization variables to obtain a convex semi-definite program (as an upper bound to the adversarial loss); while this works for their problem as minimizing upper bound also minimizes adversarial loss, there is still a gap between optimal adversarial loss and the optimal SDP solution. Also, this works specifically because of the specific non-convex structure that is similar to the MAXCUT problem. Similarly, in Wong and Kolter [2], the $l_\infty$ ball is passed through ReLU networks and a convex envelope for the resultant non-convex shape is found, again a technique specific for the $l_\infty$ ball and ReLU functions.
>
> Here we tackle a fractional structure with piece-wise linear approximation and MISOCP reformulation, which are very different with those used in these other works. A convexification is immediate in our problem if we relax the binary variables in the MISOCP to obtain a SOCP (convex problem), which unfortunately at this general level of the problem can lead to an arbitrarily large gap in the worst case.
>
> ***Baselines***: The DRO formulation for us is the max min formulation and the two timescale TT-GAD is designed precisely for such formulation even with non-convexity (two gradients in different timescale for the max and min respectively) to reach stationary points. Thus, TT-GAD was used for DRO and simple PGD for the max VR problem.
>
> The Qi et al. [3] technique also converges to a stationary point (local optima) and we observe that PGD and TT-GAD do also converge for our problems. One of our focus is global optima, thus, all these baselines provide similar solution quality outcomes when they converge.
>
> ***Theorem 1 and MISOCP bound***: The MISOCP bound shows the gap between an approximate solution instead of the optimal $\widehat{\mathbf{z}}^{\*\*}$ for the DRO formulation (the VR form) in terms of growth with $K$.
> Theorem 1 sheds light on a total performance bound as a function of the *N* number of samples of $\widehat{\mathbf{x}}$ and the *N*$_T$ samples used in training to output these $\widehat{\mathbf{x}}$'s and if the DRO problem can be solved exactly to obtain $\widehat{\mathbf{z}}^{\*\*}$. As a typical learning theory bound, the results here provides insight about the performance bound in terms of growth with *N, N*$_T$, but the bounds are typically very conservative as they hold for any true distribution (very much like PAC theory bounds) and hence it is a common practice to infer hyperparametes such as number of pieces from the typical datasets itself, as we do in our experiments. Nonetheless, the importance of the theorem and the MISOCP bound lies in identifying the terms that can reduce the approximation or performance bound.
>
> ***Computation time***: 0.37 seconds is the time taken to cluster 80,000 points into 50 clusters which is negligible compared to the runtimes of a few minutes of the MISOCP in the appendix. We also report the runtimes of baseline gradient methods here (and added to the appendix as Table 6 and 7 in revised version); as expected they are faster than solving a MISOCP but the local optima reached provides a much inferior solution quality as reported in the main paper (Table 2 in main paper, Table 4 in Appendix).
>
> |Training time (seconds) using PGA across various settings | | | | | | ||
> |-----|-----|---------|---------|---------|----|-----|-------|
> |||***MC-FCP***||||***MC-FLP***||
> ||m=7 | m=10 | m=13 || m=10 | m=12 | m=14|
> |ERM | 82.16 | 142.47 | 228.54 || 71.62 | 158.27 | 211.48|
> |1E2 | 91.24 | 147.56 | 280.44 || 81.37 | 143.44 | 230.92|
> |1E3 | 90.11 | 197.63 | 250.54 || 73.16 | 153.17  | 233.56|
> |1E4 | 83.45 | 178.12 | 320.56 || 82.84  | 167.66 | 242.28|
>
> |Training time (seconds) using TTGD across various settings | | | | | | | |
> |------|-------|--------|---------|---------|-------|------|--|
> |||***MC-FCP***||||***MC-FLP***||
> ||m=7 | m=10 | m=13 || m=10 | m=12 | m=14|
> |ERM | 44.17 | 50.31 | 62.13 || 40.11 | 45.64 | 60.63|
> |1E2 | 45.12 | 52.12 | 60.01 || 42.34 | 43.11 | 63.18|
> |1E3 | 50.11 | 43.17 | 64.32 || 45.77 | 46.23  | 63.66|
> |1E4 | 43.43 | 55.82 | 62.54 || 42.14 | 47.76 | 60.28|
>
> [1] Raghunathan et al. Certified Defenses against Adversarial Examples. ICLR 2018.
>
> [2] Wong and Kolter. Provable defenses against adversarial examples via the convex outer adversarial polytope. ICML 2018.
>
> [3] Qi et al. An Online Method for A Class of Distributionally Robust Optimization with Non-Convex Objectives. NeurIPS 2021.

---

### Official Review · Reviewer_CQhS · 2022-07-11

**Rating:** 6
**Confidence:** 2
**Soundness:** 3 good
**Presentation:** 2 fair
**Contribution:** 3 good

**Summary:**

The authors propose an approach for a distributionally robust solution to a class of non-convex sum of fractional solutions. The theoretical analysis of the solution’s global optimality is provided. To further enhance the scalability of the design, several techniques are applied. The effectiveness of the mechanism is presented in simulations.

**Questions:**

1. The work exists as an incremental contribution on, e.g., (Duchi and Namkoong, 2019, based on which the critical variance regularizer is applied. Please highlight the new and more challenging things considered.
2. In DRO, what is the specific meaning of \delta_{\xi,n}? Please clarify.
3 In Sec.4, the clustering approach is applied. As Lemma 1 only states the property among clusters, why would we still have the guarantee for each sample? Do you have any assumptions about the divergence among the cluster?


**Strengths And Weaknesses:**

Strength:
1. The paper studies an interesting topic and proposes a new algorithm for global optimum。
2. The analysis of the results is solid.



Weakness
1. The incremental contribution of the work is not clearly stated. The authors should highlight their novelty compared with the ones mentioned in the work.
2. The explanation of the theoretical results is insufficient, making the related parts hard to follow.

---

> ### Author Response · Authors · 2022-08-02
> **Thank you for the Review**
>
> We thank the reviewer for the review.
>
> ***Incremental Contribution Critique*** : Our work contribution and motivation is quite orthogonal to the work by Duchi and Namkoong. Duchi and Namkoong showed equivalence of two DRO formulations. It does not consider non-convex loss functions and scalability of solution computation is not an emphasis in that work. While we do use their DRO formulation, \emph{our focus} is on a general class of non-convex loss functions, and importantly we aim for and achieve globally optimal DRO solution for this non-convex class, and with guarantees---we show that gradient based methods do not achieve global optimality in our problems. Also, we show applications for our algorithmic contribution inspired from real world problems that are non-convex.
>
> ***Explanation*** : $\Delta_{\xi, n}$ is defined on line 96 right after its usage in the DRO equation; it is a subset of the probability simplex around the uniform distribution $\frac{\mathbf{1}}{N}$ on the data samples $\widehat{\mathbf{x}}_1 ,\ldots, \widehat{\mathbf{x}}_N$, where any point $\mathbf{p}$ is within $2\xi/N^2$ distance of $\frac{\mathbf{1}}{N}$.
>
> ***Question 3*** : Lemma 1 is not just a result only about clusters, it connects statistic (specifically, mean and variance) of the whole data to same statistic of the cluster centers. It is saying the mean $\widehat{\mbox{Mean}(F(\mathbf{z},\mathbf{x}))}$ of $F$ over all the data points is
> close to the $\widehat{\mbox{Mean}}^S(F(\mathbf{z},\mathbf{x}))$ computed using the cluster centers. Similar claim holds for the variance $\widehat{\mbox{Var}}(F(\mathbf{z},\mathbf{x}))$ using all data points and the variance $ \widehat{\mbox{Var}}^S(F(\mathbf{z}, \mathbf{x})) $ using cluster centers. We do have an assumption on the cluster radius being less than $\epsilon$ (stated in line 249 and 250). We are not very sure about what the reviewer means by ``guarantee for each sample'', but our guarantees are for the optimal solution $\widehat{\mathbf{z}}$ output using all cluster centers versus the one $\mathbf{z}^*$ that would be output using all the data without clustering (Thm 3).

---

> > ### Comment · Reviewer_CQhS · 2022-08-08
> > **Thanks for the rebuttal**
> >
> > Thank you for your response. The response addressed several concerns I had and I have raised the score.

---

> > > ### Author Response · Authors · 2022-08-09
> > > **Thanks a lot!**
> > >
> > > We thank the reviewer for the kind words and for raising the scores.

---

### Official Review · Reviewer_TpEB · 2022-07-11

**Rating:** 7
**Confidence:** 3
**Soundness:** 3 good
**Presentation:** 3 good
**Contribution:** 3 good

**Summary:**

In this paper the authors provide a reformulation for a non convex DRO problem with a fractional objective. Specifically, they focus on scalability and leverage existing results on reformulation and extend them to their setting. To achieve further scalability they apply clustering to the original data points.
They illustrate the benefits of their results on a variety of different applications and illustrate them through numerical data.

**Questions:**

Are there potential ways of extending this DRO method (which seems to primarily relay the Duchi reformulation) to other ambiguity sets?
What are the limitations of this particular ambiguity sets?


**Limitations:**

The discussion on limitations is quite limited focusing primarily on scalability. It would be good to see some discussion outside this domain as well such as about the limitations of the DRO model etc.


**Strengths And Weaknesses:**

Originality: I believe that the paper is quite original. It showcases an important way forward for the application of DRO in practical settings by focusing on problem structure and leveraging the same to allow for better scalability.
Quality: I feel the paper is sounds. The claims are well illustrated by numerical experiments as well as relevant theoretical results. The paper presents a complete picture of their work.
Clarity: I feel the paper is fine though the compactness with which some results are written makes them difficult to read.
Significance: I feel the paper takes an important step in the direction extending the applicability of DRO to more practical problems.

---

> ### Author Response · Authors · 2022-08-02
> **Thank you for the Review**
>
> We thank the reviewer for the comments and overall positive assessment. We have added a few points stated below in the conclusion of the revised main paper.
>
> ***Extending to other ambiguity sets***: Another popular ambiguity set is one based on a Wasserstein ball around the empirical distribution. Extending our analysis to Wasserstein ball is non-trivial, since a lot of work in Wasserstein based DRO rely on duality of the risk function [1, 2,  3], which also is further tractable only for specific loss functions. We believe our fractional loss is not tractable with Wasserstein ambiguity set, but this is also a promising future research direction.
>
> Further, for f-divergence based ambiguity sets (including the $\chi^2$ one that we use), one immediate limitation is the absolute continuity requirement in definition of f-divergences; in other words, the nearby distributions considered are only discrete distributions with support on the given data samples. Thus, it does not fully cover the probability space around the empirical distribution.
>
> [1] Mohajerin Esfahani, Peyman, and Daniel Kuhn. "Data-driven distributionally robust optimization using the Wasserstein metric: Performance guarantees and tractable reformulations." Mathematical Programming 171.1 (2018): 115-166.
>
> [2] Blanchet, Jose, and Karthyek Murthy. "Quantifying distributional model risk via optimal transport." Mathematics of Operations Research 44.2 (2019): 565-600.
>
> [3] Gao, Rui, Xi Chen, and Anton J. Kleywegt. "Distributional robustness and regularization in statistical learning." arXiv preprint arXiv:1712.06050 (2017).

---

### Meta-Review · Area_Chair_bsJs · 2022-08-26

**Recommendation:** Accept
**Confidence:** Less certain

**Metareview:**

This paper studies distributionally robust optimization for a class of non-convex problems motivated by applications to decision problems such as security games and facility location. The reviewers agree that the paper is novel, studies an interesting topic, and proposes non-trivial algorithms with sound analysis. Reviewer bkf3 writes that the clustering and stratified sampling scaling approaches are simple yet elegant solutions. The main common concern among all reviewers i that some sections of the paper are compactly written and somewhat difficult to read. Based on the reviewer feedback, this paper is above the bar for NeurIPS.

Below are a number of typos and clarifications that should be made in the final version of the paper:

- Lines 37, 43: It looks like some citations here need \citep instead of \citet.
- Line 51: Should “location facility” be “facility location”?
- Line 91: Missing 1/n factor (though obviously this does not affect the optimal decision variables z).
- Line 93, 96: I think \mathcal{P}_{\zeta, n} should be \mathcal{P}_{\zeta, N}. I realize this notation is borrowed from the Duchi and Namkoong paper, but in their setting the number of samples was little n.
- Paragraph starting on line 86: I think it would be worth clarifying the various distributions at play. My understanding is that there is an (currently unmentioned) distribution D over feature vectors. The distribution P is the distribution over x* corresponding to the following sampling procedure: Sample a feature vector b from D, then set x* = f*(b), where f is the Bayes optimal classifier. Next, fix a sample b_1, …, b_N from D. hat P is the empirical distribution of f(b_i), where f is a learned approximation to f*, and P* is the empirical distribution of f*(b_i).
- Theorem 1: I think it should be explicitly stated that this theorem applies when the function f is an empirical risk minimizer.
- Line 107: I think \mathcal{H} should be the function class \mathcal{F} from the previous paragraph.
- Equation (1) and line 132: variable i is used twice for different purposes (i.e., two different sum indices in equation (1) and as the index of l_i and the sum index in line 132).
- Line 141: Should this be a piecewise *constant* approximation? I.e., if we use the components of the v variable as indicators for which piece of the approximation I am in, an expression of the form dot(a,v) + b can only represent piecewise constant functions instead of piecewise linear functions. I might not be seeing the intended transformation, but nevertheless it would be good to clarify this.
- Line 174: Are the functions n and d assumed to be Lipschitz with respect to both arguments, or just x?
- Theorem 2: It might be good to clarify how the bound depends on the Lipschitz constant of the functions n and d. I assume this contributes to the number of pieces needed in the piecewise linear approximation.


**Award:**

No

---

### Decision · Program_Chairs · 2022-09-14

Accept